# VIDEO GENERATION BEYOND A SINGLE CLIP

## ABSTRACT

This work tackles the challenge of generating long videos, which entails producing videos that surpass the output length of video generation models. Due to computational constraints, video generation models are restricted to generating relatively short video clips compared to the length of real-world videos. Existing approaches employ a sliding window technique to generate long videos during inference, but this method is often restricted to homogeneous content and recurring events. To generate long videos that encompass diverse content and multiple events, we propose utilizing additional guidance to steer the video generation process. We further introduce a multi-stage approach to address this challenge, enabling us to leverage existing video generation models to produce high-quality videos within a limited time window while holistically modeling the long video based on the provided guidance. Our method complements existing video generation efforts. Extensive experiments on challenging real-world videos demonstrate the advantages of the proposed method, which outperforms the state-of-the-art by up to 9.5% in objective metrics and is preferred by users over 80% of the time. The source code and trained models will be released to the public.

## 1 INTRODUCTION

Video generation has recently attracted increasing attention. As a natural extension of image generation, most existing works treat video generation as a 3D volume prediction problem and focus on generating realistic video clips. While this paradigm has demonstrated impressive progress in a wide range of video generation tasks such as frame prediction (Srivastava et al., 2015; Finn et al., 2016; van Amersfoort et al., 2017; Oh et al., 2015; Mathieu et al., 2015), class-conditional generation, unconditional video generation (Vondrick et al., 2016; Saito et al., 2017; Tulyakov et al., 2018; Saito et al., 2020; Clark et al., 2019), etc., the length of the generated video clip is inherently constrained by computational resources. Due to the substantial computational and memory overhead incurred by state-of-the-art video generation models, they often generate video clips significantly shorter than real-world videos.

To generate long videos that match the length of real videos, existing works adopt a sliding window approach. Specifically, the model generates one video clip at a time in temporal order while taking the previously generated frames as input. The previously generated frames serve as the condition for the model to ensure that the generated video is consistent across video clips. This approach has been successfully applied to generate long videos (Ge et al., 2022). However, the results are far from satisfactory, particularly in real data domains. The synthesized videos are often limited to homogeneous videos of natural scenes or videos with recurrent human actions. In contrast, many real videos contain dynamic scenes with multiple, non-recurrent events.

The sliding window approach is sub-optimal for two reasons. First, since a generated clip relies solely on the preceding clip, it fails to capture the long-range dependencies within the video, such as the reappearance of an object that had previously left the frame. Second, it assumes that the initial frames are a sufficient condition for generating long videos. However, accurately inferring the future solely based on the initial frames is improbable (Yushchenko et al., 2019). To address these shortcomings, incorporating additional guidance to steer the generation process and holistically modeling the entire video is crucial.

In this work, we tackle the *long video generation* problem through using additional guidance as control. We formulate the problem as: given an existing video generation model, a series of guidance, and a reference frame representing the initial condition of the video, the goal is to generate a video

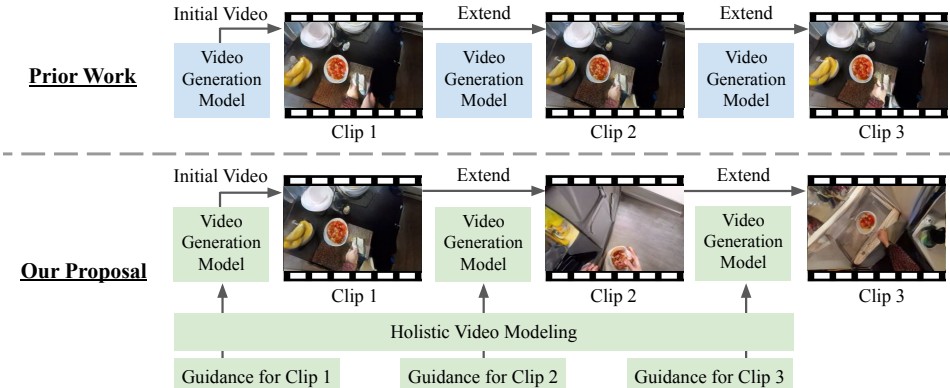

Figure 1: **Long video generation.** Existing schemes (Ge et al., 2022) generate long videos using a sliding window approach that iteratively extends the generated short video, which often leads to videos with repetitive patterns. In contrast, we explore the problem of generating long videos using additional guidance and holistically modeling the entire video, enabling the generation of videos that encompass multiple, non-repetitive yet coherent events.

covering the events specified by the guidance. The proposed problem aims to generate *long video* covering *multiple events* specified by *additional guidance* and is complementary to existing efforts that focus on generating high-quality videos within a fixed temporal window.

We employ object labels as high-level guidance, providing information about the objects that will be present in the generated video clip. Additionally, we utilize image layouts as low-level guidance, which can either be generated from the object labels or directly provided by users. While other forms of guidance are possible, object labels and layouts offer user-friendly input and naturally support interactive video manipulation such as content insertion or removal.

To solve the long video generation problem, we propose a multi-stage approach by decomposing the problem into keyframe generation followed by frame interpolation. We first predict all the keyframes jointly based on the input guidance and reference frame. These keyframes represent the starting frames of each video clip. We then generate the entire video by predicting the intermediate frames between keyframes, using the video clip generation model. See Figure 1. The multi-stage approach allows us to utilize existing video generation models that are highly optimized to generate high-quality, realistic videos within a short temporal window.

To encourage the temporal consistency of the generated videos, our core idea is to: 1) model the full video holistically to capture long-range dependencies through the keyframe generation module and 2) constrain the interaction between layout and keyframe to preserve the object coherency in the long video using layout-keyframe attention mask. Our evaluations show that the holistic keyframe modeling and the attention module help to improve the consistency throughout the video.

We conduct extensive quantitative and qualitative experiments on both real and synthetic data. In particular, we evaluate our method on the EPIC Kitchen dataset, which is challenging for video generation due to the rapid motion and complex scenes. Empirical results verify the advantage of the proposed long video generation method and show that it outperforms state-of-the-art video generation models by 9.5% on FVD. Our main contributions are as follows. First, we study the long video generation problem which aims to extend the capability of existing video generation models to generate long videos with dynamic scenes and non-recurrent events. Second, we propose a multi-stage approach for the long video generation problem. Finally, we conduct extensive evaluations to validate the efficacy of the proposed framework.

## 2 RELATED WORK

**Video synthesis.** Video prediction, class-conditional video generation and unconditional video generation have been widely studied as the sub-tasks of video generation. GAN-based methods (Vondrick et al., 2016; Saito et al., 2017; Tulyakov et al., 2018; Clark et al., 2019; Saito et al., 2020; Skorokhodov et al., 2022a; Tian et al., 2021; Brooks et al., 2022) have demonstrated early success

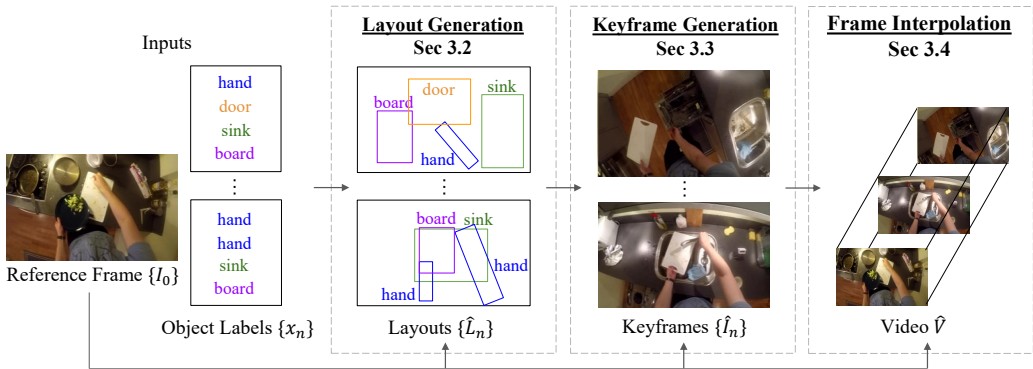

Figure 2: **Approach overview.** The proposed method consists of three stages: 1) *Layout generation* predicts a series of layouts from the object label sets. 2) *Keyframe generation* predicts a sequence of keyframes of the video from the layout sequences and the reference frame. 3) *Frame interpolation* synthesizes the intermediate frames given the keyframe sequence to obtain the complete video.

in generating short video clips, while the generation quality decreases significantly when applying to long videos. Diffusion models (Voleti et al., 2022; Yang et al., 2022; Ho et al., 2022b) are recently introduced for video generation. However, the slow sampling speed of diffusion models limits their ability to generate long videos. Auto-regressive models are first developed to synthesize raw pixels in videos (Kalchbrenner et al., 2017; Weissenborn et al., 2020; Babaeizadeh et al., 2020). Thanks to the development of vector quantization (Esser et al., 2021; van den Oord et al., 2017) and transformer (Devlin et al., 2019) models, auto-regressive methods are adapted to predict discrete tokens in the latent space (Yan et al., 2021; Moing et al., 2021; Ge et al., 2022) with impressive visual quality. In this work, we build upon recent advances of VQVAE and non-autoregressive transformer models (Chang et al., 2022; Yu et al., 2022a; Kong et al., 2022).

Despite the recent success in video generation, these models mostly focus on generating short clips (*e.g.* 16-frame videos) and are limited to synthesizing videos in specific domains (Villegas et al., 2023; Ge et al., 2022), such as human actions (Soomro et al., 2012; Siarohin et al., 2019; Bodla et al., 2021), sky timelapse (Xiong et al., 2018), robotics videos (Ebert et al., 2017). Most recently, a few text-to-video models (Wu et al., 2021; 2022; Hong et al., 2023; Singer et al., 2023; Ho et al., 2022a; Blattmann et al., 2023) are developed to generate videos given natural language inputs. However, these models are limited to videos of single scenes without a meaningful storyline. In contrast, our work aims to generate videos with diverse content and novel events. In addition, we focus on generating videos beyond the length of a single clip and any existing video clip generation models can be adopted as a component in our pipeline. There are few works exploring video generation conditioned on input guidance at multiple timesteps (Bar et al., 2021; Yu et al., 2022b; Ge et al., 2022; Harvey et al., 2022; Yin et al., 2023) while limited to synthetic environments. In contrast, our work aims to deal with real-world datasets. On the other hand, unconditional video generation (Brooks et al., 2022; Skorokhodov et al., 2022b) successfully generates long videos but is not applicable to our conditional scenario.

**Story visualization and image manipulation.** Story visualization (Li et al., 2019; Song et al., 2020; Maharana et al., 2021; Maharana & Bansal, 2021) focuses on synthesizing a sequence of images that visualize a story of multiple sentences. GeNeVA (El-Nouby et al., 2019; Fu et al., 2020; Zhenhuan Liu, 2020; Cong et al., 2022) is a conditional text-to-image generation task developed on CoDraw (Kim et al., 2019) dataset. It studies the problem of constructing a scene iteratively based on a sequence of descriptions. However, these two lines of work are limited to experiments on synthetic and cartoon data. These approaches focus on generating a few frames of a visual story instead of videos. In contrast, we focus on experiments on real-world data and generating videos.

## 3    APPROACH

In this section, we introduce the proposed method for long video generation. We first give an overview of the approach and then describe the three main components.

## 3.1 OVERVIEW

Given 1) a series of $N$ input guidance, and 2) a reference frame, $I_0$ as input, our goal is to synthesize a video $V$ that covers the content provided in the guidance while starting from the initial frame $I_0$. The reference frame provides the same initial condition as existing video prediction tasks. We use sets of object labels $\{x_1, x_2, \cdots, x_N\}$ to serve as the input guidance. Each object label set $x_n$ contains $k^n$ object labels which indicates the objects that will appear in the video. Note that $k^n$ may vary across different timesteps $n$.

To tackle the long video generation problem, our core idea is to model the entire video jointly through keyframe generation and utilize existing models to generate high-quality videos within a short temporal window. We achieve these by 1) generating a series of $N$ keyframes in the video given the object label sets, each keyframe presents the initial and final frames of the short video clips, and 2) predicting the intermediate frames between the keyframes to obtain the complete video. Furthermore, we introduce an additional layout generation stage that explicitly predicts the 2D layouts of the keyframes to reduce the difficulty of keyframe generation. See Figure 2.

The proposed approach offers several noteworthy advantages. First, it facilitates holistic video modeling over an extended time window by representing the video in an abstract intermediate form. The layout generation process aids in preserving the 2D frame structure, which can easily deteriorate over time during long video generation. Second, it enables the utilization of various video generation models to generate intermediate video frames, providing flexibility in video generation. Lastly, it empowers users to manipulate the generated videos through layout adjustments. See Section A.3 for examples of such manipulations. The following sections provide a detailed description of the three stages of our approach.

## 3.2 LAYOUT GENERATION

We first generate a series of layouts from the object label sets to explicitly constrain the keyframe generation. Given a series of $N$ object label sets $\{x_n\}$, and a reference frame $I_0$, we synthesize a series of layouts $\{L_n\}$, which represents the layouts of the $N$ keyframes in the video. Since the reference first frame $I_0$ is given, we assume $L_0$ is known and can be used as a reference layout. We define the layout $L$ as a set of bounding boxes with variable length $k^n$, $i.e.$ $\{b_1, b_2, \cdots, b_{k^n}\}$. The bounding box attributes include its object label, x-coordinate, y-coordinate, width and height, $i.e.$ $b = \{c, x, y, w, h\}$.

Our layout generator is a transformer-based model. We first apply tokenization to encode the layouts $\{L_n\}$ and the object label set $\{x_n\}$ into discrete tokens. We then train the model to predict the layout tokens given the object label tokens as input. We flatten the input series of object label sets $\{x_n\}$ and layouts $\{L_n\}$ into token sequences where the values of the bounding box attributes are simply discretized by uniform quantization. To predict the layout series $\{\hat{L}_n\}$, the tokens of the reference layout $L_0$ and the object labels $c$ are given and the other attributes of the bounding boxes $\{x, y, w, h\}$ are replaced with a [MASK] token. The transformer model is trained to predict the masked bounding box attributes for $\{\hat{L}_n\}$. Our layout generation module predicts multiple layouts jointly at the same time so that it preserves the temporal consistency of the generated layout sequence.

## 3.3 KEYFRAME GENERATION

Next, we generate a sequence of keyframes from the predicted layout sequences. Given a reference first frame $I_0$ and a series of layouts $\{\hat{L}_n\}$ either given by the users or synthesized in the previous stage, we aim to generate a sequence of keyframes $\{\hat{I}_n\}$. We use a transformer-based model as our keyframe generator. A VQVAE (Esser et al., 2021) is used to encode the images $I$ into visual tokens $e$, and a bidirectional transformer model is used to predict the tokens for the target keyframes. Finally, the decoder of VQVAE is used to map the predicted keyframe tokens $\hat{e}$ back into raw images $\hat{I}$. Specifically, the keyframe sequence $\{I_n\}$ are transformed into tokens $i.e.$ $\{e_n\}$. At training time, the tokens of layouts and keyframes are concatenated as $s = \{L_0, \hat{L}_1, \cdots, \hat{L}_N, e_0, e_1, \cdots, e_N\}$ in dataset $\mathcal{D}$. We randomly replace the tokens in the sequence with the [MASK] token and obtain the masked sequence $s_M$. The transformer model is trained to predict the masked tokens $s_t, t \in M$ by

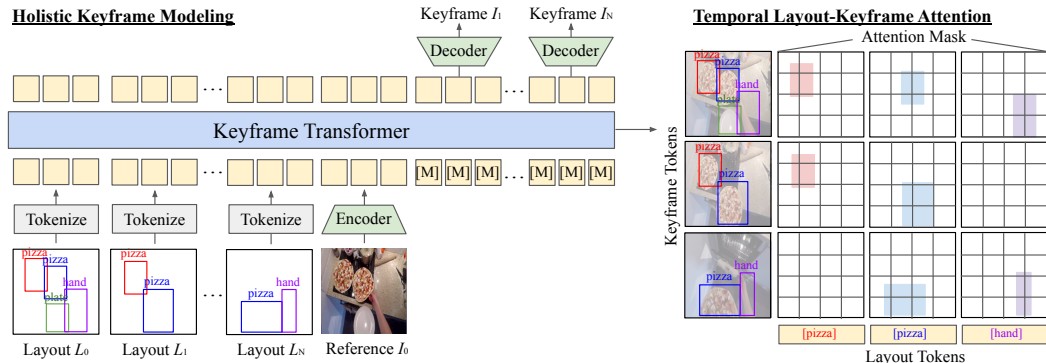

Figure 3: **Keyframe generation.** Our key ideas are: 1) Holistic keyframe modeling: the transformer model predicts multiple keyframes jointly to maintain the coherency given the layouts and the reference frame as input. 2) Temporal layout-keyframe attention: the attention mask between irrelevant keyframe and layout tokens is set as zero to encourage the model to focus on relevant tokens where the tokens within a bounding box and the layout tokens represent the same object in the sequence.

minimizing the negative log-likelihood:

$$\mathcal{L} = - \mathop{\mathbb{E}}_{s \in \mathcal{D}} \sum_{t \in M} \log p(s_t | s_M) \tag{1}$$

At test time, the tokens of the layout sequence and the first frame $\{L_0, \hat{L}_1, \cdots, \hat{L}_N, e_0\}$ are given, and the model predicts the tokens of the following keyframes $\{\hat{e}_n\}$, which is reconstructed to the target keyframes $\{\hat{I}_n\}$ by the decoder of VQGAN. See Figure 3. Our keyframe generator predicts all the frames jointly. This provides a holistic model for the entire video. As shown in the experiment, our design improves the consistency across keyframes and the coherency of the video.

**Temporal layout-keyframe attention.** To further improve the quality and coherency of keyframe generation, we propose a temporal layout-keyframe attention mask. As our input to the keyframe generator is a concatenation of the layout and keyframe tokens, each token is attended to every other token without considering the relationship between the layout and the keyframe. However, this may be redundant as the visual tokens within a specific bounding box in the keyframe ($\hat{e}_n$) are highly related to the layout tokens of the corresponding bounding box in $\hat{L}_n$. In addition, as we generate a series of keyframes jointly, the tokens of each bounding box in a single layout $\hat{L}_n$ are correlated with every relevant visual token in all the keyframes $\{e_n\}$, as long as the bounding box represents the same instance in the sequence. We encode this prior knowledge into our model through the temporal layout-keyframe attention mask, where we set the irrelevant attention as zero in order to guide the model to focus on relevant tokens. This prior knowledge helps the model to 1) respect the relation between layout and keyframe to improve their alignment and 2) constrain the generation of the same instance in the sequence to be correlated to improve the coherency.

## 3.4 FRAME INTERPOLATION

Finally, given the reference frame $I_0$ and a sequence of generated keyframes $\{\hat{I}_n\}$ as inputs, we can adopt any existing video clip generation model (Ge et al., 2022; Yu et al., 2022a) to generate the video between the keyframes. Specifically, we use a transformer-based model as our frame interpolation model. The model predicts the intermediate frames given the initial and final keyframe as input. We first convert the video into discrete tokens using a 3D-VQVAE model. A transformer model is then used to predict the masked video tokens of intermediate frames. Finally, the interpolated video tokens are mapped back to the raw videos by the 3D-VQVAE decoder.

During inference time, given two consecutive keyframes $\hat{I}_{n-1}$ and $\hat{I}_n$, the model predicts the video token sequences that connect between the two keyframes, *i.e.* $\hat{z}_n$. $N$ clips of video tokens $\{\hat{z}_n\}$ are predicted and concatenated as a sequence to be mapped to the long video by the 3D decoder. Note that our frame interpolation requires video generation between keyframes with a long temporal distance, while existing works (Huang et al., 2022; Bao et al., 2019) are limited to interpolate adjacent frames with small motion. See Section D.1 for further comparison.

Table 1: **Quantitative results of video generation on EPIC Kitchen.** LPIPS is averaged across all frames, AP is evaluated using the $45^{th}$ frame. Our method consistently outperforms state-of-the-art video generation methods on video, image and detection metrics.

| Methods | FVD ↓ | LPIPS ↓ | AP ↑ |
|---|---|---|---|
| TATS | 737.6 | 0.615 | 0.072 |
| StyleGAN-V | 581.8 | 0.550 | 0.074 |
| MAGVIT (frame pred.) | 291.4 | 0.475 | 0.133 |
| MAGVIT (class cond.) | 285.6 | 0.469 | 0.140 |
| Ours | **258.4** | **0.421** | **0.192** |
| Ours (GT layouts) | 214.7 | 0.346 | 0.390 |
| Ours (GT keyframes) | 174.1 | 0.242 | 0.451 |

Table 2: **Quantitative results on nuScenes.**

| Methods | FVD ↓ | LPIPS ↓ | AP ↑ |
|---|---|---|---|
| MAGVIT (frame pred.) | 171.1 | 0.478 | 0.044 |
| Ours | **158.9** | **0.448** | **0.059** |
| Ours (GT layouts) | 106.0 | 0.384 | 0.137 |
| Ours (GT keyframes) | 84.5 | 0.306 | 0.219 |

Table 3: **User study.** We report the percentage of raters that prefer our method per video quality and reproduction of the ground truth videos.

| Methods | Quality | Reproduction |
|---|---|---|
| Ours *vs.*MAGVIT (frame pred.) | 76.3% | 82.1% |
| Ours *vs.*MAGVIT (class cond.) | 68.4% | 66.7% |

## 4 EXPERIMENTS

We validate our approach on both challenging real-world videos and synthetic data. We first evaluate the video generation results by comparing with state-of-the-art methods. Next, we verify our model design through ablation study.

**Dataset.** We validate our method on three datasets in different domains. 1) **EPIC Kitchen** (Damen et al., 2022) consists of egocentric videos of kitchen activities. Compared with commonly used datasets for video generation research, *e.g.* UCF (Soomro et al., 2012), Kinetics (Kay et al., 2017), and BAIR (Ebert et al., 2017), EPIC Kitchen videos contain complex and dynamic scenes. The foreground objects move in and out of the camera field-of-view, and the background scenes and camera viewpoint change frequently, *e.g.* moving to different rooms. To synthesize such video, the video generation model needs to generate multiple, non-recurrent events within a short time window, *i.e.* multiple actions performed by the subject. 2) **nuScenes** (Caesar et al., 2019) consists of driving videos including both the motion of objects *e.g.* moving cars or people, and the dynamics caused by the egocentric motion *e.g.* background scenes change over time. 3) **CoDraw** (Kim et al., 2019) is a synthetic dataset consisting of sequences of images. While it is not a video dataset, it contains diverse object classes and provides complete annotation for the object labels and layouts. We use CoDraw to evaluate our keyframe generation. See Section C.1 for additional details.

**Evaluation metrics.** **FVD** assesses the quality of generated videos by measuring whether the distribution of generated videos is close to that of real videos in the feature space. **FID** assesses the quality of generated images similar to **FVD**. **LPIPS** assesses the similarity between the generated frames and the ground truth video frames. **AP and F1** assess the semantic accuracy of generated videos by measuring the correctness of the generated objects. We measure AP by comparing the detected objects between the generated and ground truth frames. See Section B.1 for details.

**Implementation details.** We leverage an off-the-shelf segmentation model (Cheng et al., 2021), not specifically trained on our dataset, to automatically extract sparse labels. This automated process eliminates the need for manual annotation and can be applied to arbitrary data. In addition to the foreground objects, the dynamic background scenes are described by "stuff" labels such as wall, floor and table. Each video sequence contains 64 frames, and the videos are generated at $128{\times}128$. We sample one keyframe every 16 frames in the 64-frame clips so that the keyframe sequence contains $N = 4$ synthesized keyframes at $256{\times}256$. See Section C.2 for additional information.

### 4.1 VIDEO GENERATION

We evaluate video generation on the EPIC Kitchen and the nuScenes datasets. The goal is to verify the effectiveness of additional guidance and holistic video modeling in long video generation.

**Baselines.** We compare with the following state-of-the-art video generation methods:

- MAGVIT (frame pred.) (Yu et al., 2022a): given the reference frame as input, we apply MAGVIT to generate a 16-frame clip. We then take the last predicted frames as input to iteratively generate the entire video. This is the standard sliding window approach for long video generation.

- MAGVIT (class cond.): we condition MAGVIT on both the reference frame and the object label. This extends the sliding window approach to take additional guidance similar to our method.
- StyleGAN-V (Skorokhodov et al., 2022b): we train their unconditional model to generate a 64-frame video, and apply GAN projection method to condition the model on the reference frame.
- TATS (Ge et al., 2022): similar to MAGVIT, we generate a 16-frame clip and use sliding window.
- Ours (GT layouts) and Ours (GT keyframes): to understand how the quality of layout and keyframe generation affects the video generation results, we compare with two variants of our method that take the ground truth layouts and keyframes as inputs.

**Quantitative results.** The results are in Table 1 and Table 2. Our method obtains substantial, *i.e.* ∼9% and ∼7% improvements upon baselines on the video metric. The generated videos are closer to the ground truth as evaluated by the image metric, and the detection score validates that our model generates videos that contain the set of objects as specified by the input guidance. The results verify the effectiveness of the proposed approach. Note that MAGVIT (class cond.) performs better than MAGVIT (frame pred.), which shows the benefit of additional guidance. Our approach further improves MAGVIT (class cond.), which suggests that both the additional guidance and the holistic video modeling provided by our approach are helpful for long video generation. On the other hand, Ours (GT layouts) and Ours (GT keyframes) significantly improve the output video quality. The results suggest that there is significant room for improvement in long video generation given the very same video generation model. It verifies that optimizing video generation for a single clip (*i.e.* 16 frames) alone is not sufficient for solving the long video generation problem. Our proposed method for generating video beyond a single clip, which is orthogonal to existing efforts, is essential. Note that our multi-stage approach allows users to manually improve the intermediate representations, *e.g.* provide more detailed layouts, which allows further improvement for the generated videos in an interactive generation setting.

**User study.** We conduct a user study to augment the quantitative evaluation. In the study, we present two videos generated by different methods together with the ground truth video and ask the raters 1) **Quality**: which video has the better visual quality, and 2) **Reproduction**: which video better reproduces the content of the ground truth video. We conduct the study with 40 videos and 11 participants. The results are in Table 3, which is consistent with the quantitative results and further validates the clearly visible gain of our method compared to the baselines.

**Qualitative results.** Figure 4 shows the per-frame LPIPS score. X-axis shows the frame number, ranging from 0 to 64. Y-axis shows the LPIPS of a specific frame averaging over all test videos. The generated image quality degraded rapidly in MAGVIT, especially when we generate videos beyond the length of the training clip (*i.e.* 16 frames). In contrast, our approach experiences a slower quality degradation, which further verifies its benefit in generating video beyond the training clip.

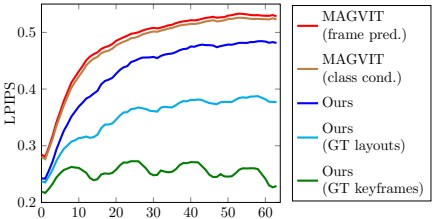

Figure 4: **Per-frame video results.** Our method slows down quality degradation.

Figure 5 and Figure 6 show the qualitative results. TATS, StyleGAN-V and MAGVIT tend to generate video with homogeneous contents, and the quality degrades for long sequences. On the other hand, our method generates video with scene change (wall to table), object deletion (plate at the bottom) and object insertion (left/right hand), showing the ability of our model to generate videos with multiple events. We further show that Ours (GT layouts) generates videos with the layouts close to the upper bound results *i.e.* Ours (GT keyframes). On the other hand, Ours generates videos that match the label sets but with different layouts. The results validate the ability of our model to generate videos that satisfy different levels of the input guidance *i.e.* object label sets or (more constrained) layouts. We show good results for the challenging long video generation problem where the objects move in and out of camera views with large motion and changing scenes.

## 4.2 KEYFRAMES GENERATION

Next, we evaluate the performance of our keyframe generation model. The goal is to verify 1) the importance of generating all the keyframes jointly, and 2) the importance of additional guidance for generating content across a large temporal window.

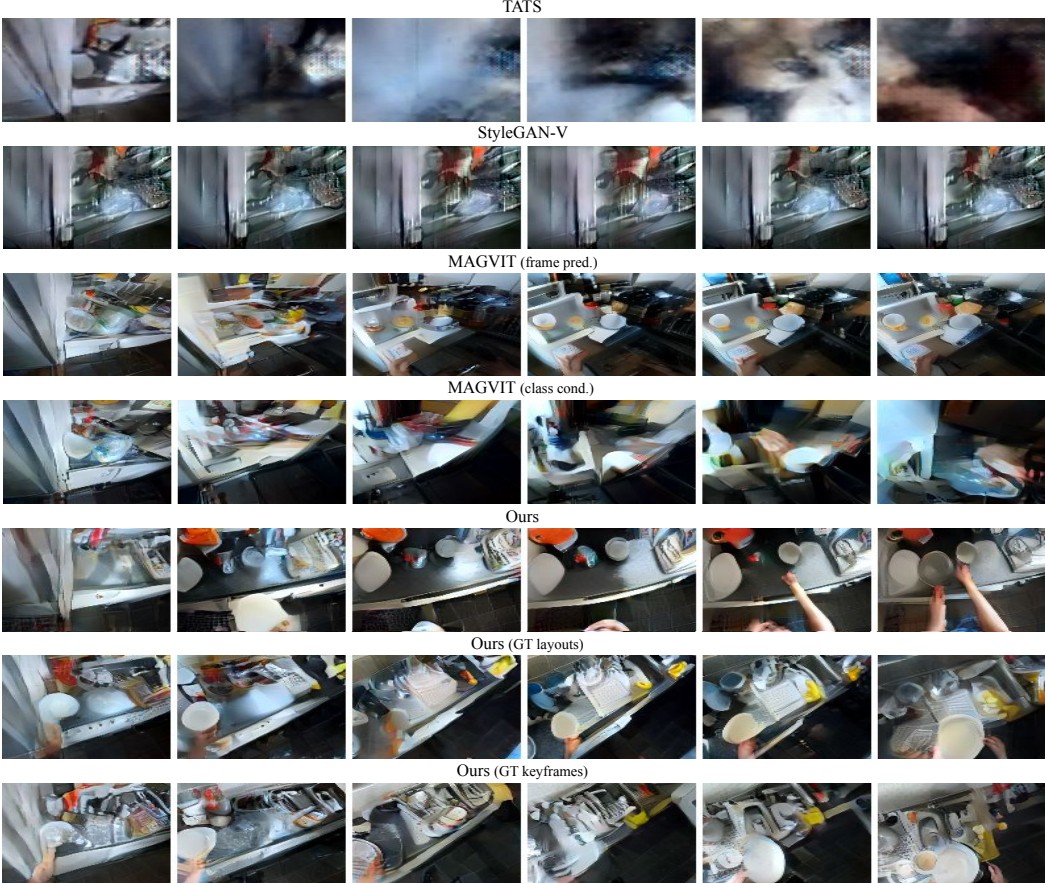

Figure 5: **Qualitative results of video generation on EPIC Kitchen.** TATS, StyleGAN-V and MAGVIT predict videos with quality degradation for long sequences. Our method generates long videos with meaningful events *e.g.* scene change, object insertion and deletion.

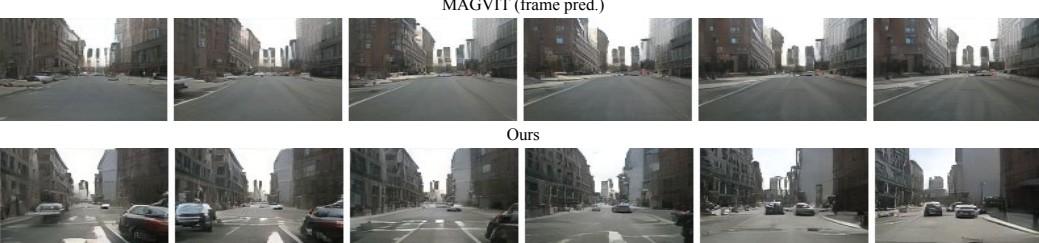

Figure 6: **Qualitative results of video generation on nuScenes.** MAGVIT predicts relatively static videos and repetitive patterns when inferring long videos that are beyond the output length of the model, while our method generates dynamic and non-recurrent long videos.

**Baselines.** We compare the following baselines and variants of our method:

- MaskGIT (Chang et al., 2022): the model takes the reference as input and iteratively predicts the next keyframe. This model represents keyframe generation without input guidance.
- HCSS (Jahn et al., 2021): the model takes a single layout as input and generates a single keyframe. This model represents independent keyframe generation without full video modeling.
- Ours: our keyframe generation given the predicted layouts as inputs.
- Ours (GT): our keyframe generation using the ground truth layouts (upper bound).
- Ours single (GT): iterative approach that predicts keyframes conditioning on the previous keyframe and the ground truth layout, representing generation without full video modeling.

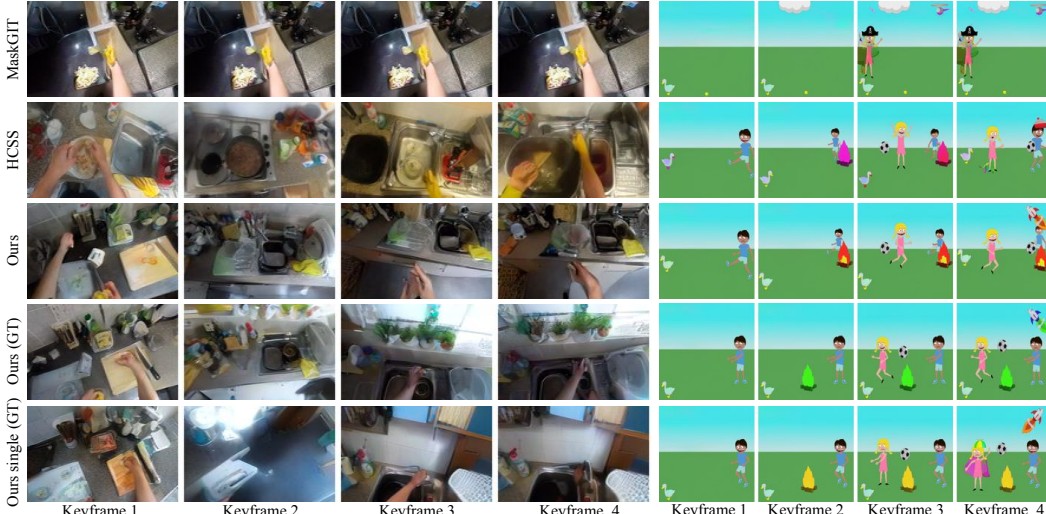

Figure 7: **Qualitative results of keyframe generation on EPIC Kitchen and CoDraw.** MaskGIT predicts similar content, and HCSS fails to generate consistent frames. The superior performance of our model shows the importance of providing additional input guidance and holistic video modeling.

**Quantitative results.** The results are in Table 4. Our method consistently outperforms HCSS on both real and synthetic data, which verifies the importance of joint prediction for all keyframes. Our method also performs better than MaskGIT except for LPIPS in the synthetic dataset. We observed that MaskGIT tends to predict repetitive keyframes with little changes across frames which implies that the video will remain static and is not suitable for video generation.

The F1 score shows that our method can generate better keyframes than MaskGIT, which verifies the importance of guidance. We also compare different variants of our method. In particular, the superior performance of Ours (GT) over Ours single (GT) further verifies that a model that considers the entire video jointly leads to better keyframe generation. See Section B.1.

Table 4: **Quantitative results of keyframe generation.** Metrics are averaged on keyframes.

| Methods | CoDraw | | | EPIC Kitchen | |
|---|---|---|---|---|---|
| | FID ↓ | LPIPS ↓ | F1 ↑ | FID ↓ | LPIPS ↓ |
| MaskGIT | 10.8 | **0.304** | 61.66 | 46.9 | 0.633 |
| HCSS | 9.8 | 0.425 | 81.55 | 50.2 | 0.653 |
| Ours | **7.4** | 0.325 | **90.04** | **29.9** | **0.548** |
| Ours (GT) | 3.9 | 0.106 | 94.68 | 24.2 | 0.416 |
| Ours single (GT) | 4.6 | 0.156 | 91.33 | 27.5 | 0.480 |

**Qualitative results.** Figure 7 shows the qualitative results. As mentioned before, MaskGIT tends to generate keyframes with similar content, which shows the importance of providing input guidance at multiple timesteps. HCSS fails to generate consistent results across the keyframes *e.g.* different styles of the sink. Similarly, we can see the iterative approach Ours single (GT) fails to generate consistent keyframes compared to Ours (GT). The examples clearly demonstrate the importance of joint modeling for the entire video.

**Ablation study of temporal layout-keyframe attention.** In Table 5, we report the result of removing the layout-keyframe attention mask so that each layout token is attended to each keyframe token, denoted as w/o mask. With the proposed attention mask, our model generates keyframes that better align with the reference frame and contain more details, achieving better FID and LPIPS.

Table 5: **Ablation study of layout-keyframe attention.**

| Methods | FID ↓ | LPIPS ↓ |
|---|---|---|
| Ours w/o mask | 34.8 | 0.575 |
| Ours | **29.9** | **0.548** |

## 5 CONCLUSIONS

We tackle the problem of long video generation which aims to generate videos beyond the output length of video generation models. We show that the existing sliding window approach is suboptimal, and there is significant room for improvement using the same video generation model. To improve long video generation, we propose to use additional guidance to control the generation process. We further propose a multi-stage approach which utilizes existing video generation models while capturing long-range dependency within the video. Empirical results validate our model design and show favorable results over state-of-the-art video generation methods.

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

# A  ADDITIONAL QUALITATIVE RESULTS

## A.1  VIDEO GENERATION

We provide a qualitative comparison of the generated videos. We demonstrate the videos synthesized by our method, StyleGAN-V, MAGVIT (frame pred.), MAGVIT (class cond.) on the EPIC Kitchen dataset and the nuScenes dataset in Figure 8, Figure 9, Figure 10 and Figure 11. We also compare with two variants of our method that take the ground truth layouts and keyframes as inputs, *i.e.* Ours (GT layouts) and Ours (GT keyframes). More video results are attached in the zip file.

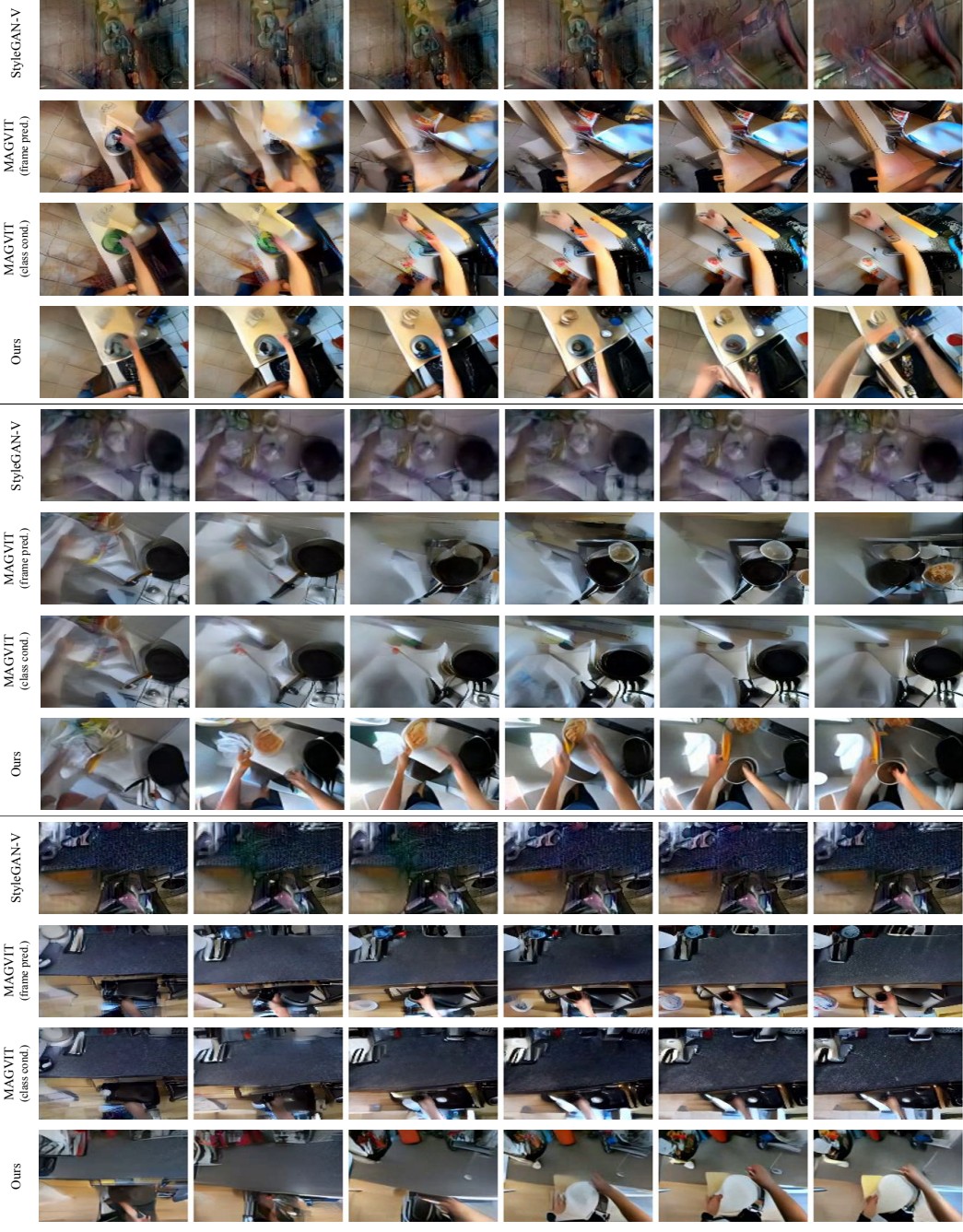

Figure 8: Additional results of video generation on the EPIC Kitchen dataset.

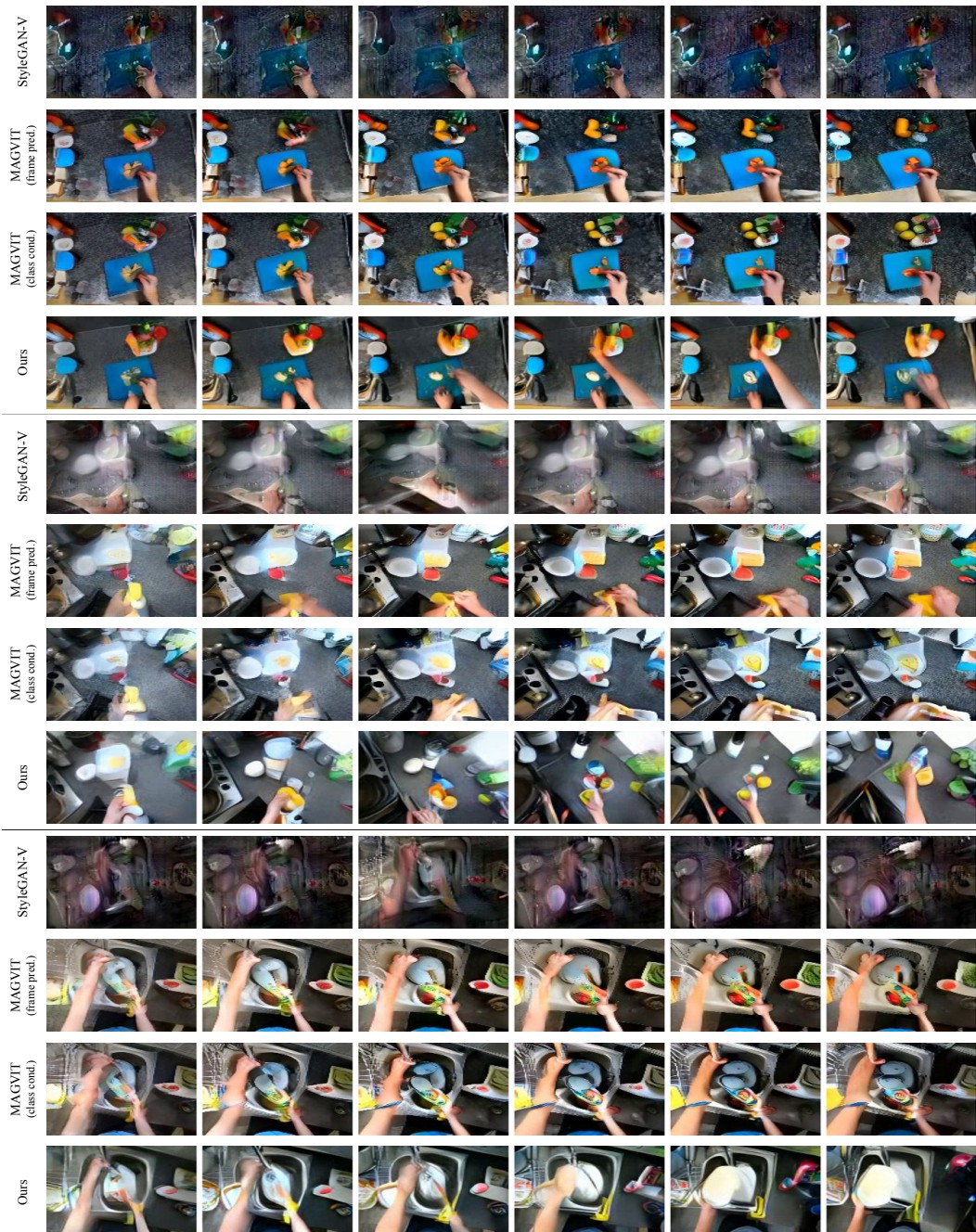

Figure 9: Additional results of video generation on the EPIC Kitchen dataset.

StyleGAN-V and MAGVIT tend to predict a relatively static video with repetitive motions, and the quality degrades for long sequences. On the other hand, our method generates video with background change, object deletion and object insertion, showing the ability of our model to generate videos with multiple events. We have shown good results in the challenging long video generation problem on real-world datasets.

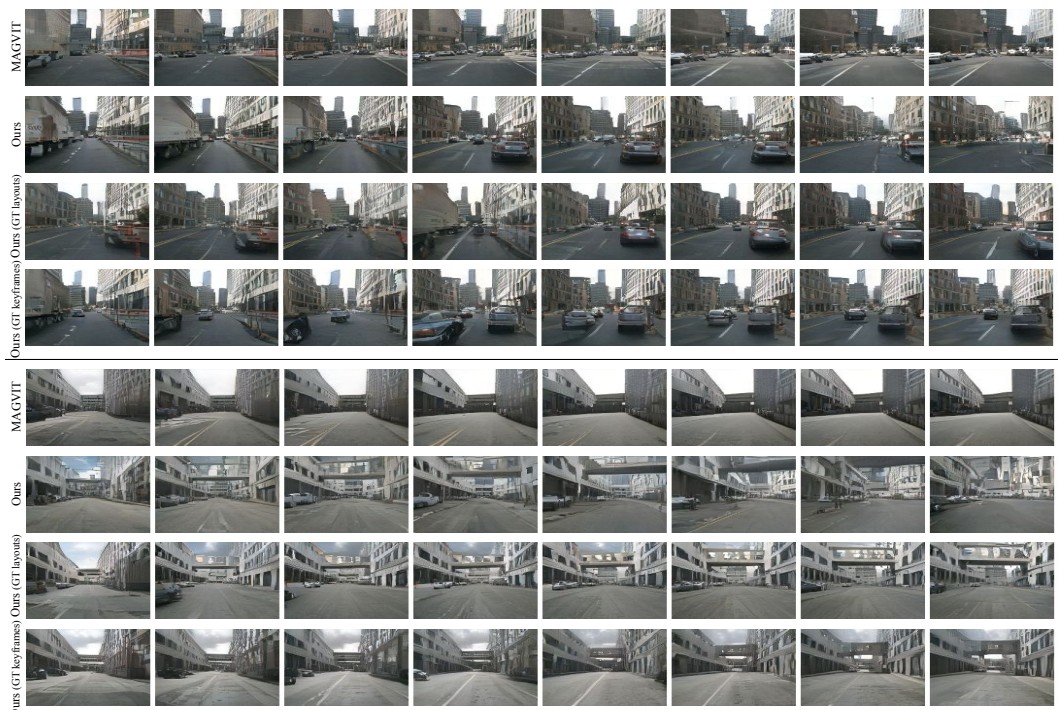

Figure 10: Additional results of video generation on the nuScenes dataset.

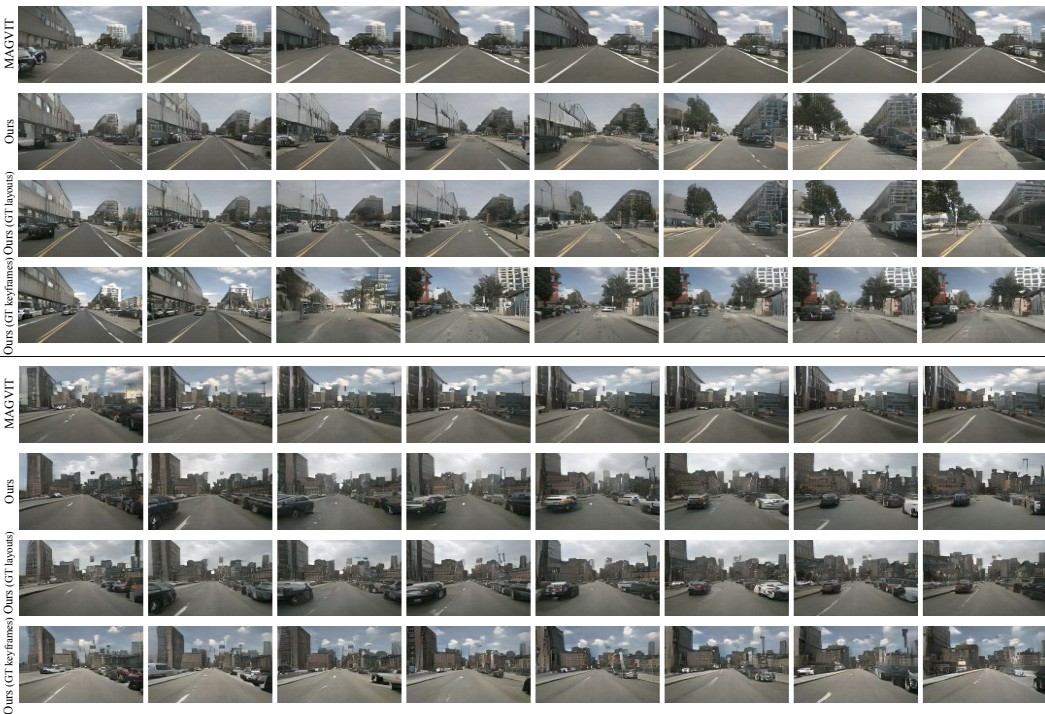

Figure 11: Additional results of video generation on the nuScenes dataset.

## A.2 KEYFRAME GENERATION

We include additional comparison of keyframe generation on the CoDraw dataset and the EPIC Kitchen dataset in Figure 12 and Figure 13. We show the keyframes generated by our method, MaskGIT (Chang et al., 2022), HCSS (Jahn et al., 2021). We also include the results of Ours (GT), where the keyframe generation uses the ground truth layouts as inputs and Ours single (GT) where the keyframes are predicted iteratively conditioning on the previous keyframe and ground truth layouts.

MaskGIT tends to generate keyframes with similar content, which shows the importance of providing input guidance at multiple timesteps. Comparing HCSS and our method, we can see that HCSS fails to generate consistent results across the keyframes. Similarly, when we compare Ours (GT) with Ours single (GT), we can see the iterative approach fails to generate consistent keyframes. The examples clearly demonstrate the importance of joint modeling for the entire video.

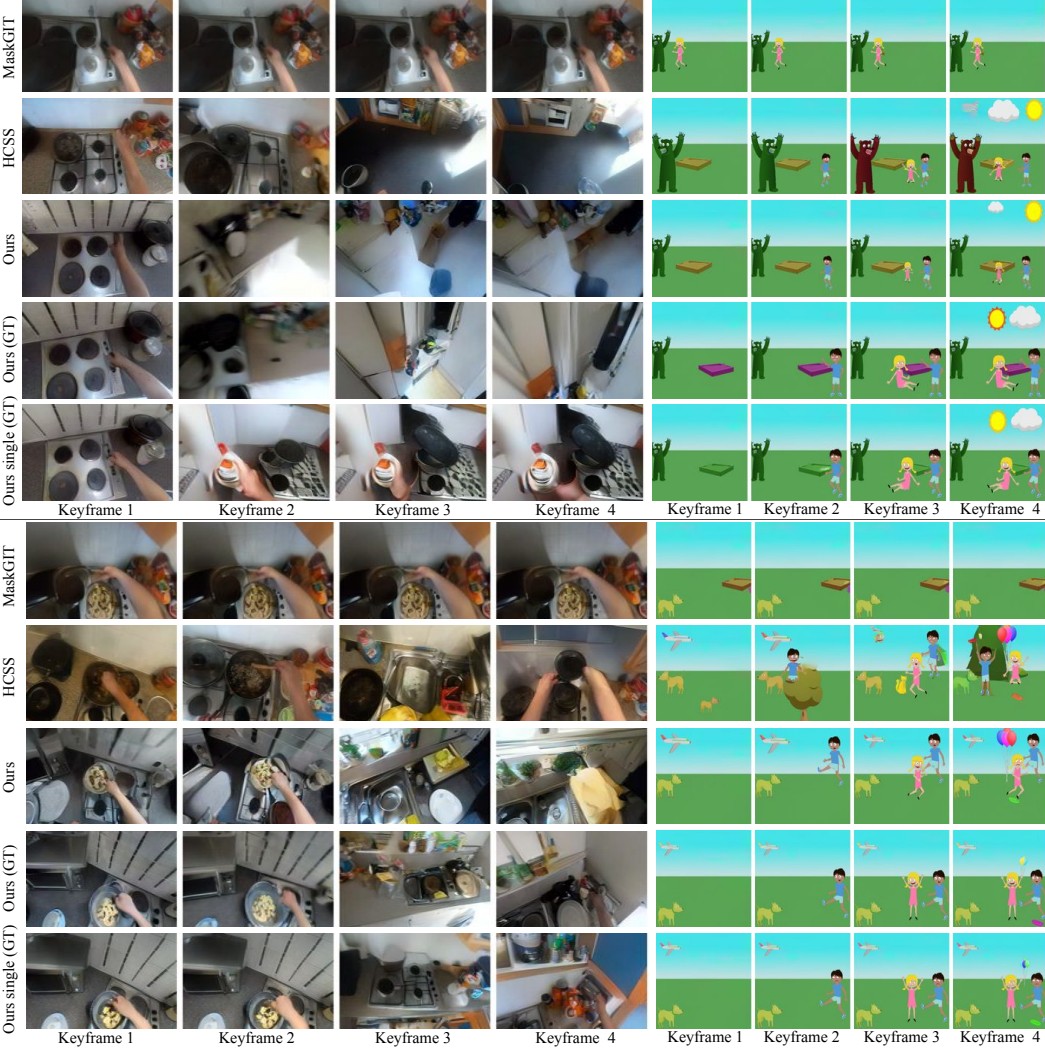

Figure 12: Additional results of keyframe generation.

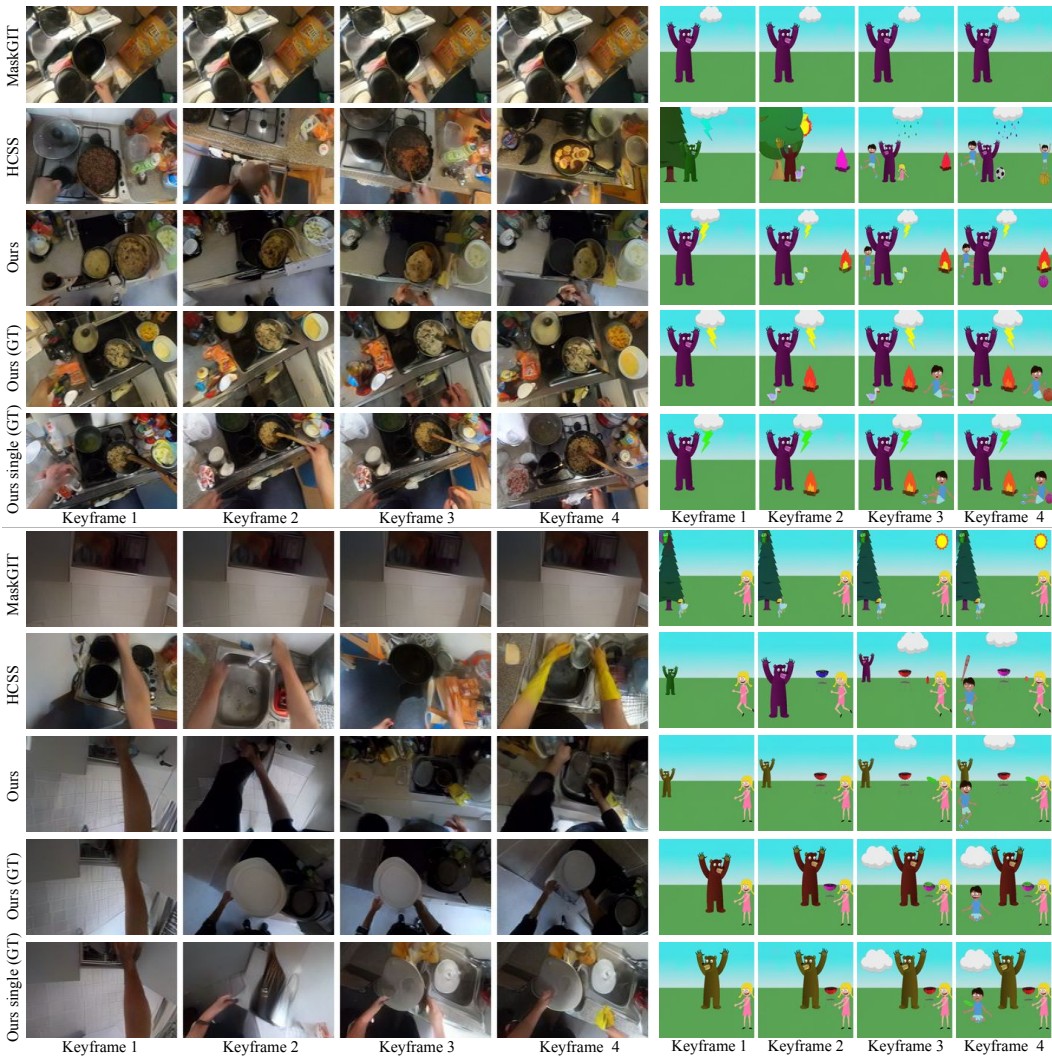

Figure 13: Additional results of keyframe generation.

## A.3 VIDEO MANIPULATION

Our layout generation module brings an additional benefit that the users can manipulate the generated videos by editing the layouts such as content insertion or removal. Figure 14 shows that our method allows users to generate different videos by sampling different layouts from the model. It also allows video manipulation through layout, where the user may remove an object, change the size and position of the objects, *etc*.

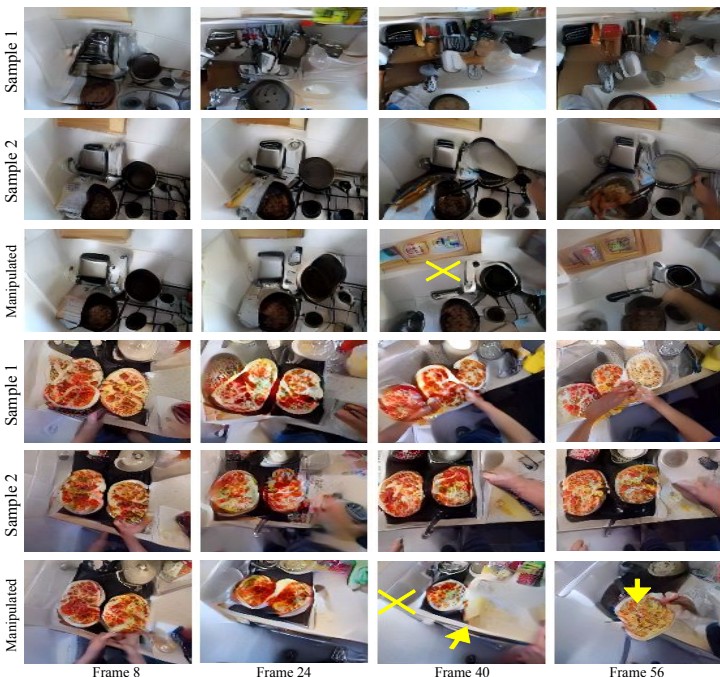

Figure 14: **Video manipulation through layouts.** Users can generate different videos by sampling different layouts, as shown in the first two rows. Users can even manipulate the videos by editing the layouts. The yellow cross in the third row shows object deletion, and the yellow arrow in the last row shows object movement.

# B ADDITIONAL QUANTITATIVE RESULTS

## B.1 KEYFRAME GENERATION

We evaluate the performance of our keyframe generation model using additional metrics PSNR and SSIM. The results are in Table 6. The PSNR and SSIM scores are consistent with LPIPS. Our method consistently outperforms HCSS on both real and synthetic data, which verifies the importance of joint prediction for all keyframes. Our method also performs better than MaskGIT except for the metrics of similarity with ground truth on the synthetic CoDraw dataset. However, we observe that MaskGIT tends to predict repetitive keyframes with little changes across frames, which is not suitable for video generation. We provide further quantitative evaluation in Table 7.

Table 6: **Quantitative results for keyframe generation.** The metrics are averaged across keyframes. We include additional metrics PSNR and SSIM. The results are consistent with LPIPS.

| Methods | CoDraw | | | | EPIC Kitchen | | | |
|---|---|---|---|---|---|---|---|---|
| | FID ↓ | PSNR ↑ | SSIM ↑ | LPIPS ↓ | FID ↓ | PSNR ↑ | SSIM ↑ | LPIPS ↓ |
| MaskGIT (Chang et al., 2022) | 10.8 | 17.822 | 0.846 | 0.304 | 46.9 | 10.471 | 0.251 | 0.633 |
| HCSS (Jahn et al., 2021) | 9.8 | 14.534 | 0.748 | 0.425 | 50.2 | 10.121 | 0.235 | 0.653 |
| Ours | 7.4 | 16.952 | 0.809 | 0.325 | 34.8 | 11.541 | 0.268 | 0.575 |
| Ours (GT) | 3.9 | 22.114 | 0.889 | 0.106 | 24.2 | 14.158 | 0.367 | 0.416 |
| Ours single (GT) | 4.6 | 20.233 | 0.860 | 0.156 | 27.5 | 12.902 | 0.320 | 0.480 |

The standard image quality metrics such as FID and LPIPS cannot evaluate whether a generated keyframe contains the set of objects given in the object label set. Following GeNeVA (El-Nouby et al., 2019), we train an object detector and localizer on our CoDraw data. We apply the object detector to the generated keyframes and the ground truth keyframes and report the object detection metrics: precision, recall and F1 scores. In addition, we report the relational similarity as in (El-Nouby et al., 2019). It constructs the graph where the vertices are objects and the edges are the positional relations. The relational similarity compares the similarity of the graph built by the generated keyframes and the ground truth keyframes.

We observe that our results are significantly better than MaskGIT (Chang et al., 2022) and HCSS (Jahn et al., 2021), which verifies a model that considers the entire video jointly leads to better keyframe generation and validates the importance of the input guidance. As discussed in Table 6, although MaskGIT obtains better LPIPS scores on the CoDraw dataset, it gets much lower object detection metrics compared to our model. The results validate that our method generates keyframes with correct objects compared to MaskGIT.

Table 7: **Object detection metrics for keyframe generation.** The metrics are averaged across keyframes. Our method achieves significantly better scores than baselines.

| Methods | Precision ↑ | Recall ↑ | F1 ↑ | Rsim ↑ |
|---|---|---|---|---|
| MaskGIT (Chang et al., 2022) | 72.40 | 58.00 | 61.66 | 52.86 |
| HCSS (Jahn et al., 2021) | 82.87 | 81.76 | 81.55 | 68.39 |
| Ours | 93.24 | 88.19 | 90.04 | 75.32 |
| Ours (GT) | 96.00 | 94.10 | 94.68 | 91.64 |
| Ours single (GT) | 92.35 | 91.27 | 91.33 | 87.38 |

### B.2 LAYOUT GENERATION

We compare the performance of layout generation to BLT (Kong et al., 2022), where their model takes a single object label set as input and generates a single layout. BLT represents independent layout generation without full video modeling.

We evaluate the performance following BLT using the below metrics:

- IOU, Overlap: measures the absolute quality with the intuition that a better layout contains objects without much overlap.
- Similarity: computes the distance between ground truth layouts and generated layouts.

Table 8 shows that our method that jointly predicts a sequence of layouts achieves better metrics than BLT where each of the layouts in the series is generated independently from a single object label set. Our model preserves the temporal consistency of the layout sequences while BLT fails.

Table 8: **Quantitative results for layout generation.** The metrics are average across keyframes. Our method obtains better scores than baseline BLT.

| Methods | CoDraw | | | EPIC Kitchen | | |
|---|---|---|---|---|---|---|
| | IOU↓ | Overlap↓ | Sim↑ | IOU↓ | Overlap↓ | Sim↑ |
| BLT (Kong et al., 2022) | 0.171 | 0.092 | 0.073 | 0.758 | 1.796 | 0.124 |
| Ours | 0.117 | 0.060 | 0.084 | 0.684 | 1.341 | 0.145 |

## C IMPLEMENTATION DETAILS

### C.1 DATASET

**EPIC Kitchen.** The dataset contains 20M frames extracted from 700 videos with a total length of 100 hours. The videos have variable lengths and cover a long period. We follow the original train and test split and cut the videos into 64-frame sequences. We first re-sample the sequences with double the frame rate, so that each 64-frame sequence covers a 5-second video with the five keyframes sampled equidistantly.

Specifically, to train the frame interpolation module, we create short training sequences of 16 frames which cover 1.28 seconds (25 fps). To train the layout generation and keyframe generation modules, we sample one frame every 16 frames and form the 5-frame keyframe sequences that cover 64 frames which are around 5.12 seconds of the video. Therefore, our model generates four 16-frame video clips of 1.28 seconds between the five keyframes and finally creates a 64-frame video covering 5.12 seconds. With the preprocessing, we create 276k sequences for training and 3k sequences of non-overlapping frames for testing.

Since there are no object labels and layout annotations in the EPIC Kitchen dataset, we use Mask-Former (Cheng et al., 2021) to extract the semantic map for each frame and convert them into the object labels and bounding boxes, which serve as the input guidance and ground truth layout. We use 133 object labels as predicted by the MaskFormer model. We further apply temporal smoothing to clean up the object labels and remove small objects that cover less than 2% of the image. Finally, the connected components extracted from the semantic segmentation maps are used as the pseudo ground truth of our object labels and layouts.

**nuScenes.** We follow the original train split and select 600 videos as the test split. We use the same frame sampling strategy as EPIC Kitchen.

**CoDraw.** The dataset contains 10k sequences of the scene manipulation process. Although the sequences have variable lengths, we sample every consecutive 5 frames to form 67k training sequences and 4k test sequences. The dataset contains 58 different objects with the same appearance for each object. To analyze the temporal consistency between the generated keyframes, we extend the original CoDraw data by creating 6 different appearances for each clip object class using color harmonization code and re-render the data using the original layouts. The object instances have the same appearance in the same sequence while having different appearances in other sequences. In addition, the object can be resized and horizontally flipped in one sequence. These changes allow us to evaluate whether the models preserve the consistency of object appearance when predicting the keyframe sequences.

## C.2 TRAINING

**Layout generation.** We first apply tokenization to encode the object label sets $\{x_n\}$ into discrete tokens. We then learn a transformer model that predicts the layout tokens given the object label sets as input. We use a transformer model with 4 layers, 8 attention heads, 512 embedding dimensions, and 2048 hidden dimensions. We use Adam optimizer and train the transformer models on $4 \times 4$ TPU with a batch size of 64 and peak learning rate $5 \times 10^{-4}$ for 100k iterations.

We set the maximum number of objects in the layout to 14. For the CoDraw dataset, we use the original layout representation $b = \{c, x, y, w, h\}$. Each object in the layout is encoded into 5 tokens, which results in a 70-token layout and the layout sequence including $5 \times 70 = 350$ tokens. If the object label set has less than 14 objects, we pad the sequences with padding tokens. At test time, the first 70 tokens and the class tokens $c$ are given, and the model predicts the bounding box attributes $x, y, w, h$. We use uniform quantization to encode bounding box attributes into integer numbers between 0 and 31.

We use a different layout representation for the EPIC Kitchen and the nuScenes datasets. The objects in real-world videos are not aligned with a rectangular bounding box in general, and there are many background pixels included if one uses bounding boxes as the layout representation. We use the extreme point representation (Zhou et al., 2019) where four points, *i.e.* (top-most, left-most, bottom-most, right-most) are used to represent the location and size of an object. The extreme point representation provides more compact boundaries for the objects and preserves the object shape and orientation information. Specifically, we use $b = \{c, x_{top}, y_{top}, x_{left}, y_{left}, c, x_{bottom}, y_{bottom}, x_{right}, y_{right}\}$. Each object in the layout is encoded into 10 tokens, which results in a 140-token layout and the layout sequence including $5 \times 140 = 700$ tokens. At test time, the first 140 tokens and the class tokens $c$ are given, and the model predicts the other extreme point attributes.

**Keyframe generation.** We convert the keyframe generation into tokenization and sequence prediction. We use a VQVAE (Esser et al., 2021) model to encode the raw images into discrete visual

tokens, and a bidirectional transformer model is used to predict the masked image tokens *i.e.* target keyframes. Finally, the decoder of VQVAE is used to map the visual token into raw images.

We train the model to generate keyframes at $256 \times 256$ resolution. The VQVAE model encodes an image into $16 \times 16$ tokens with codebook size 1024. Similar to the layout encoding process above, one layout is translated and padded into a sequence of 128 tokens. Therefore, the transformer model takes a sequence of $(16 \times 16 + 128) \times 5 = 1920$ with $(16 \times 16) \times 4 = 1024$ token replaced with [MASK] as input, and the model predicts the masked tokens.

We use a transformer model with 24 layers, 8 attention heads, 768 embedding dimensions, and 3072 hidden dimensions. We use Adam optimizer and train the transformer models on $4 \times 4$ TPU with a batch size of 128 and peak learning rate $10^{-4}$ for 400k iterations. We utilize the VQVAE model pre-trained on ImageNet (Chang et al., 2022) and finetune it for 1200k iterations on our datasets.

**Frame interpolation.** We use a transformer model to generate intermediate frames similar to the video interpolation task of MAGVIT (Yu et al., 2022a). The model takes the initial and final frames of the video as input. It first converts the input frames into discrete video tokens using 3D-VQVAE. A transformer model is then used to predict the tokens of intermediate frames. Finally, the interpolated video tokens are mapped back to the raw videos by the 3D decoder.

We train the model to generate 16-frame videos at $128 \times 128$ resolution with stride 2. The 3D-VQVAE model encodes a 16-frame video into $4 \times 16 \times 16$ tokens using the codebook size 1024. We use MAGVIT 3D-VQ (B) with 41M parameters. The transformer model predicts a sequence of length 1024. We use MAGVIT (B) with 128M parameters. Both models are trained on batch size 128 using Adam optimizer and a peak learning rate $10^{-4}$ with linear warm-up and cosine decay for 100k iterations. We implement the model in JAX and the training is conducted on $8 \times 8$ TPU-v2 chips.

At test time, the transformer model is applied $N$ times to predict the tokens between two keyframes which serve as the initial and final frame of the video clips. Specifically, we repeatedly pad the initial frame and the final frame 8 times to form a $16 \times 128 \times 128$ video, which is encoded to $4 \times 16 \times 16 = 1024$ tokens. The transformer model is then used to re-generate the corresponding 1024 tokens. We obtain $N$ sequences and the tokens sequences are concatenated to sequences of length $N \times 1024$ and reshaped to $4N \times 16 \times 16$. We input the reshaped tokens into the 3D-VQVAE decoder to reconstruct $16N \times 128 \times 128$ video. $N$ is set as 4 in our experiments.

### C.3 BASELINES

**MAGVIT.** We train MAGVIT using the same transformer architecture and training strategy as ours for fair comparison. We train the model to generate 16-frame videos at $128 \times 128$ resolution. We use MAGVIT 3D-VQ (B) and MAGVIT (B). Both models are trained on batch size 128 using Adam optimizer and a peak learning rate $10^{-4}$ with linear warm-up and cosine decay for 100k iterations. We implement the model in JAX and the training is conducted on $8 \times 8$ TPU-v2 chips.

**TATS, StyleGAN-V.** We train the model using their released code and training strategy.

**MaskGIT, HCSS.** We train MaskGIT and HCSS using the same transformer architecture and training strategy as ours for fair comparison. We train the model to generate keyframes at $256 \times 256$ resolution. The VQVAE model encodes an image into $16 \times 16$ tokens with codebook size 1024. One layout is translated and padded into a sequence of 128 tokens. We use Adam optimizer and train the transformer models on $4 \times 4$ TPU with a batch size of 128 and peak learning rate $10^{-4}$ for 400k iterations.

## D DISCUSSIONS

### D.1 COMPARISON TO FLOW-BASED INTERPOLATION

We use transformer-based model (Yu et al., 2022a) as our frame interpolation module (Section 3.4) instead of using flow-based interpolation models as we study a different problem. The flow-based models (Niklaus & Liu, 2020; Huang et al., 2022; Bao et al., 2019) interpolate adjacent frames within a short temporal window (0.04sec, 25fps to 100fps, $4 \times$ frames) while our task requires interpolating

keyframes with a long temporal distance (0.64sec, 1.56fps to 25fps, 16×frames). Their standard benchmarks (UCF and Vimeo) contain limited and holistic motion, *e.g.* <6 pixels in a 128×128 image, and our task tackles large and local motion with removed/added objects between frames. As shown in Figure 15, our method (bottom) creates the motion of hand and cut board moving in the frame while RIFE (top) cannot handle large motion and generates blurry results.

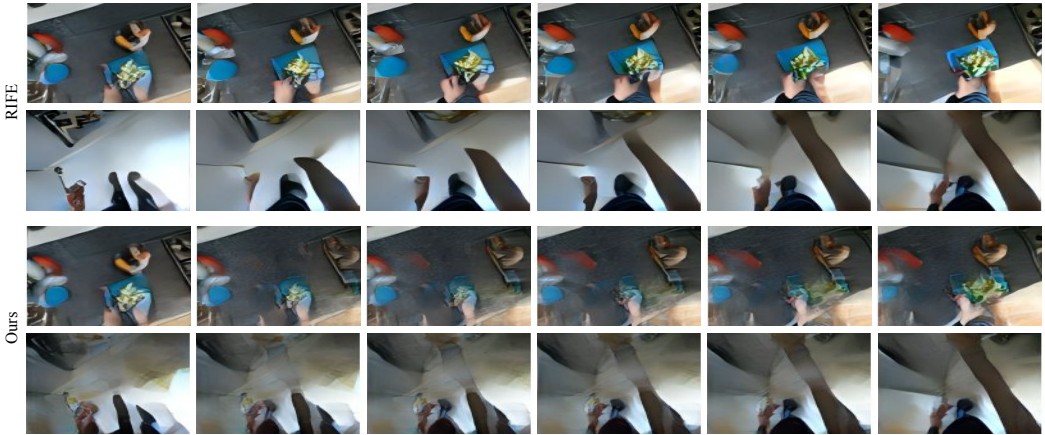

Figure 15: Comparison to flow-based interpolation.

## D.2 COMPARISON TO LONGVIDEOGAN

EPIC Kitchen contains complex scene structures consisting of multiple objects, *e.g.* motion caused by both camera and object movement, changing scenes (moving to different rooms), and the videos consist of multiple events, *i.e.* multiple actions performed by the subject. Additional guidances at multiple timesteps are therefore essential to describe the videos. In contrast, motions in the videos from LongVideoGAN (Brooks et al., 2022) are dominated by (forward) camera movement with a few objects in the same outdoor scene, and the videos contain monotonous events which can be described more compactly. We train LongVideoGAN on our data and show that it cannot generate high-quality videos as shown in Figure 16. Note that LongVideoGAN is an unconditional generation model and cannot be compared with ours directly.

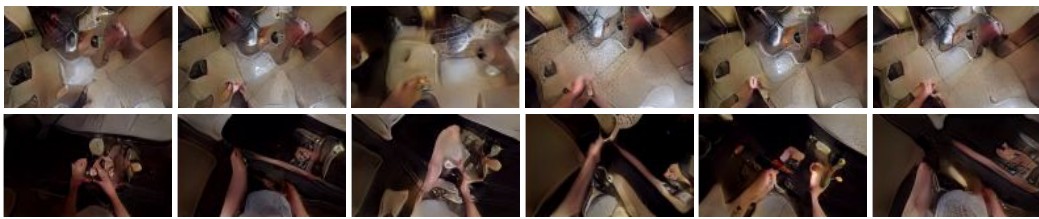

Figure 16: **Comparison to LongVideoGAN.** We show two random videos generated from (Brooks et al., 2022). They are mainly holistic contents without considering objects.

## E LIMITATIONS AND FUTURE WORK

We discuss the limitations of our method as follows. First, the output quality of our method is limited by the frame interpolation module. Specifically, the output videos do not accurately cover the transition of the hand action, which contains complex non-rigid motions. In addition, the details of the objects *e.g.* hand and containers still need further improvement to capture. Second, our multi-stage approach successfully utilizes the powerful frame interpolation module of existing work to reduce the difficulty of synthesizing long videos to achieve good quality. However, the current training strategy does not allow different stages to correct the errors of other training stages since

the SoTA frame interpolation modules are all large in size and cannot be combined with other stages. We plan to explore an end-to-end learning approach in future work. Third, our evaluation metrics do not fully reflect the improvement of our model on the long video generation task, because FVD is biased towards image quality and will be good if a model generates good images but degenerate motion and the image metrics cannot fully reflect all the possibility of generated videos. Better metrics for long video generation should be developed in future work.

