# OpenReview forum: "Video Generation Beyond a Single Clip"
_ICLR.cc/2024/Conference — ICLR 2024 Conference Withdrawn Submission_

### Official Review · Reviewer_cBdr · 2023-10-20

**Soundness:** 3 good
**Presentation:** 3 good
**Contribution:** 2 fair
**Rating:** 6
**Confidence:** 4

**Summary:**

The paper proposed a multi-stage approach to generate long videos that encompass diverse content and multiple events, which utilizing additional guidance to steer the whole video generation process. These additional guidance includes object labels, box, and layout.

Specifically, this paper first predict all the keyframes jointly based on the input guidance. These keyframes represent the starting
frames of each video clip. Then generating the entire video by predicting the intermediate frames between keyframes, using the video clip generation model.

The proposed method outperforms state-of-the-art video generation models by 9.5% on FVD.

**Strengths:**

1. The approach of initially predicting keyframes for different clips in a long video and then interpolating seems reasonable. Indeed, to some extent, it can address the issue of repetitive content in long videos.

2. Constrain the interaction between layout and keyframe to preserve the object coherency in the long video using layout-keyframe attention mask.

3. Achieved good performance on two public datasets.

**Weaknesses:**

1. This work is meaningful and has demonstrated good performance on some mainstream datasets, marking a significant improvement compared to previous efforts. However, in the era of large models and datasets, video diffusion models have shown more promising performance with features such as high resolution and more reasonable motion [1][2][3]. If the authors could further expand their training dataset to enhance performance and generalization capabilities, it would make this work even more noteworthy. Of course, the reviewer understands the substantial effort required for such tasks and does not necessarily expect the authors to provide additional experiments or proofs.

2. Furthermore, the reviewer recommends that the authors provide more visual demonstrations from other domains, not limited to cooking and driving scenarios. This would offer a more comprehensive showcase of long video generation applications across various scenes.

3. Figure 4 presents a comparison of relevant quality degradation. An interesting observation is that using GT layouts appears significantly better than layouts predicted by the transformer. Layouts seem to play a crucial role in this method. The reviewer recommends that the authors provide visualizations of layouts, especially in bad cases. This can assist readers in understanding the performance of the layout component.

[1] Wang, Yaohui, Xinyuan Chen, Xin Ma, Shangchen Zhou, Ziqi Huang, Yi Wang, Ceyuan Yang et al. "LAVIE: High-Quality Video Generation with Cascaded Latent Diffusion Models." arXiv preprint arXiv:2309.15103 (2023).
[2] He, Yingqing, Menghan Xia, Haoxin Chen, Xiaodong Cun, Yuan Gong, Jinbo Xing, Yong Zhang et al. "Animate-a-story: Storytelling with retrieval-augmented video generation." arXiv preprint arXiv:2307.06940 (2023).
[3] Ho, Jonathan, William Chan, Chitwan Saharia, Jay Whang, Ruiqi Gao, Alexey Gritsenko, Diederik P. Kingma et al. "Imagen video: High definition video generation with diffusion models." arXiv preprint arXiv:2210.02303 (2022).

**Questions:**

1. User study. The paper conducted the study with 40 videos and 11 participants. Perhaps the authors could provide a brief introduction to the origin and data distribution of these 40 videos, along with any relevant references if they align with previous works. Additionally, clarification on the participant demographics would be helpful, such as whether they are members of the author's team or hired participants. Providing this information would enhance the overall understanding of the study.

---

### Official Review · Reviewer_e4d3 · 2023-10-26

**Soundness:** 2 fair
**Presentation:** 3 good
**Contribution:** 2 fair
**Rating:** 5
**Confidence:** 4

**Summary:**

This study addresses the challenge of generating longer videos than current video generation models can produce due to computational limitations. Instead of the common sliding window technique, which is limited to consistent content, the authors propose a multi-stage approach with additional guidance to produce diverse, eventful long videos. Their method, which enhances existing video generation models, has proven superior in real-world tests, outperforming competitors by 9.5% and being favored by users 80% of the time. The main contributions of this paper are: A new method to study the long video generation problem by guidances and a layout module for keyframes, which is essential to guide the generation of subsequent frames.

**Strengths:**

1. Approach: The paper introduces a new multi-stage method for generating long videos. The approach employs additional guidance to enable high-quality video generation within a limited time window.
2. Presentation: The paper is well-organized and provides supplementary materials for readers to understand the proposed method and its components.

**Weaknesses:**

1. It lacks a comprehensive explanation on maintaining temporal consistency, especially when generating intermediate frames under keyframes.
2. The evaluation section is not robust, with limited results in the general video domain.
3. There are concerns regarding the model's capacity for fine-grained generation and conditional control, with VQGAN's performance being questionable in these areas.
4. The paper misses discussions or comparisons with relevant works like "Towards Smooth Video Composition" and "AnimateDiff-Unlimited Context Length mode."
5. In the visualization experiments, the variety of scene categories is limited, raising doubts about the model's adaptability to diverse or open scenes.


[R1] Zhang, Qihang, et al. "Towards Smooth Video Composition." ICLR 2023.

[R2] Guo, Yuwei, et al. "Animatediff: Animate your personalized text-to-image diffusion models without specific tuning." arXiv 2023.

**Questions:**

See above

**Details Of Ethics Concerns:**

The generated content may contain some concept bias.

---

### Official Review · Reviewer_V5GZ · 2023-10-29

**Soundness:** 3 good
**Presentation:** 4 excellent
**Contribution:** 3 good
**Rating:** 5
**Confidence:** 5

**Summary:**

The paper addresses the challenge of generating long videos by proposing a multi-stage approach that incorporates additional guidance to steer the video generation process. The authors argue that existing methods, which utilize a sliding window technique, are restricted to homogeneous content and recurring events. The key contributions include the introduction of a holistic video modeling technique through key-frame generation and frame interpolation, enabling the generation of diverse and non-repetitive long videos. The method demonstrates superiority over existing approaches in both objective metrics, user study and qualitative results.

**Strengths:**

1. S1. Long video generation is a notorious task. It requires a lot of computational overhead and more elaborate model design. There is currently no universally recognized effective solution. The paper focuses on generating long videos with diverse content and non-repeating events, which importantly addresses a key limitation of existing video generation models.
2. S2. The incorporation of additional auxiliary information, including labels and layouts, has proven to be an effective solution to generate long videos in this paper.
3. S3. Quality: The proposed method outperforms state-of-the-art models in objective metrics, showcasing the high quality of generated videos.

**Weaknesses:**

1. W1. Although a substantial number of quantitative and qualitative experiments are provided in this paper, they are all conducted on the 64-frame video. It is doubtful whether a 64-frame video is considered as the long video. The absence of experimental results for extended video scenarios casts doubt on the scalability of the proposed method. Relying solely on 64-frame videos undermines the comprehensive evaluation of the method's capabilities in long video generation. More results, such as $\ge$ 128-frame videos, are needed.
2. W2. While the integration of additional auxiliary conditions proves effective in long video generation, concerns arise regarding the generalizability of this approach. The method rely on the robust labels and layouts as auxiliary conditions may not be universally applicable. Not all videos can extract labels and layouts, like rivers and starry skies. Have the authors considered other types of auxiliary conditions?
3. W3. Video diffusion models are a popular topic. It is suggested to discuss more about the advantages of the proposed method over recent diffusion-based video generation methods. Considering that the SOTA method with diffusion model can generate high-quality videos (100+ frames of 512x512 pixels), what will be the advantages of the proposed method?
4. W4. This paper lacks clarity concerning the computational cost of the proposed method, particularly in terms of reasoning cost and efficiency. A more explicit delineation of these computational aspects is imperative for a comprehensive understanding of the method's practical implications and resource requirements.
5. W5. Reference missing:
- Zhang et al., “Towards Smooth Video Composition”, ICLR 2023
- Yoo et al., “Towards End-to-End Generative Modeling of Long Videos with Memory-Efficient Bidirectional Transformers”, CVPR 2023

**Questions:**

Please refer to the weaknesses, especially W1, W2, and W3.

---

### Official Review · Reviewer_ZyRs · 2023-10-31

**Soundness:** 2 fair
**Presentation:** 3 good
**Contribution:** 2 fair
**Rating:** 3
**Confidence:** 5

**Summary:**

This paper proposes a long video generation method by using the layout to guide the generation process. It breaks the generation into multiple stages, which first generate a sequence of layouts and use these layouts as conditions to generate keyframes of videos. Finally, it adopts frame interpolation to generate the intermediate frames. The authors evaluate their method on three datasets and achieve improved performance.

**Strengths:**

1. The proposed method achieves improved performance on three datasets
2. The authors provide a thorough analysis of related work.

**Weaknesses:**

1. The idea of first generating keyframes and then interpolating temporally for long video generation has been proposed by many previous works, like "Make-A-Video: Text-to-video Generation Without Text-Video Data", "CogVideo: Large-scale Pretraining for Text-to-Video Generation via Transformers" and "Align your Latents: High-Resolution Video Synthesis with Latent Diffusion Models".
2. The authors propose to first generate the layout of video content, i.e. bounding boxes with class labels.  However these things are not readily accessible, and they require a substantial amount of manual annotation. Hence, it's not suitable for scaling up.
3. Using simple domain-specific datasets for evaluation cannot adequately validate the proposed method's effectiveness. Even for these simple cases, the visualization result does not seem impressive and demonstrates severe artifacts.

**Questions:**

How to generalize the proposed method to open-domain video generation given that the layout annotations of a lot of training video data are missing?